# Clinical Application of Rapid Upper Limb Assessment and Nordic Musculoskeletal Questionnaire in Work-Related Musculoskeletal Disorders: A Bibliometric Study

**DOI:** 10.3390/ijerph20031932

**Published:** 2023-01-20

**Authors:** Venkata Nagaraj Kakaraparthi, Karthik Vishwanathan, Bhavana Gadhavi, Ravi Shankar Reddy, Jaya Shanker Tedla, Mastour Saeed Alshahrani, Snehil Dixit, Kumar Gular, Gaffar Sarwar Zaman, Vamsi Krishna Gannamaneni, Mohamed Sherif Sirajudeen, Gopal Nambi

**Affiliations:** 1Department of Medical Rehabilitation Sciences, College of Applied Medical Sciences, King Khalid University, Abha 61421, Saudi Arabia; 2Faculty of Physiotherapy, Parul University, Vadodara 391760, Gujarat, India; 3Department of Orthopaedics, Parul Institute of Medical Sciences and Research, Parul University, Vadodara 391760, Gujarat, India; 4Department of Clinical Laboratory Sciences, College of Applied Medical Sciences, King Khalid University, Abha 61421, Saudi Arabia; 5Department of Physiotherapy, Hail University, Hail 55255, Saudi Arabia; 6Department of Physical Therapy and Health Rehabilitation, College of Applied Medical Sciences, Majmaah University, Al Majmaah 11952, Saudi Arabia; 7Department of Health and Rehabilitation Sciences, College of Applied Medical Sciences, Prince Sattam bin Abdulaziz University, Al Kharj 16273, Saudi Arabia

**Keywords:** work-related musculoskeletal disorders, bibliometrics, citation analysis, top cited articles

## Abstract

Assessment of work-related musculoskeletal disorders (WMSDs) using the Rapid Upper Limb Assessment (RULA) and the Nordic Musculoskeletal Questionnaire (NMQ) has become widely accepted and reported in the literature. The objectives of this study are to (1) recognize and describe the topmost 50 cited scientific articles in WMSDs using the RULA and NMQ and (2) explore the factors that contribute to making an article influential. In this bibliometric study, we used the Web of Science and MEDLINE databases to identify the top 50 cited articles published from 1993 to 2022. The data collected were the title of the journal, number of citations, year of publication, type of the study, institution where the work was conducted, level of evidence, contribution of primary authors, and country of origin of the work. Our results showed that the top 50 cited articles were published between 1980 and 2010. The 2000s was the most valuable decade. Regarding journals, the Work journal had the highest number of articles concerning the use of RULA and NMQ in healthcare professionals. The maximum number of citations regarding RULA occurred in the Journal of Robotic Surgery (*n* = 50) and the maximum for NMQ occurred in the Journal of Safety Research (*n* = 106). Most articles originated from the United States, followed by England and the Netherlands. Eight authors had two publications published in the top 50 list. The majority of the topmost cited research articles were cross-sectional studies. Most of these studies were level III evidence. The bibliometric analysis from this study provides insights to researchers to choose the most appropriate and influential journal for submitting work on WMSDs.

## 1. Introduction

Work-related musculoskeletal disorders are considered to be a major issue worldwide [1]. Awkward non-physiological postures while working, hereditary causes, aging, and psychological issues could be the responsible factors causing work-related musculoskeletal disorders (WMSDs). These disorders include damage to various components of the locomotor system such as muscles, joints, nerves, and other connective tissues [2]. Healthcare specialists with prolonged interaction with patients had the greatest percentage of WMSDs [3]. WMSDs are recognized as a major occupational hazard in various countries [4,5,6,7] since there is a considerable economic burden imposed by these conditions [8,9].

The use of a validated, uniform evaluation tool is recommended in order to diagnose and devise effective interventions to tackle WMSDs [10]. Two such important evaluation tools that have been utilized for many years are the Rapid Upper Limb Assessment (RULA) [11] and the Nordic Musculoskeletal Questionnaire (NMQ) [12]. Both these tools demonstrated greater reliability in evaluating ergonomic hazards in various healthcare practitioners [4,13,14].

The RULA was developed by Corlett and McAtamney in 1993, primarily as an observational tool. It was utilized to assess the person’s susceptibility to loads due to the posture of the upper extremity and spine and the assistance of the lower extremities required to cope with the extra load during one’s work. The overall RULA score ranges from one to seven. Higher scores suggest a greater likelihood of a musculoskeletal problem. RULA scores either equal to or greater than five necessitate a change in posture during work [11]. 

The Nordic Musculoskeletal Questionnaire (NMQ) was developed by Kuorinka et al. in 1987. It is an easy, well-defined questionnaire comprising a body map describing nine functional sites revealing both sides’ upper limbs, lower limbs, upper back, and lower back. It incorporates questions on symptoms felt by the individuals in the previous 12 months and the past seven days, along with constraints in activity levels in the previous 12 months. The responses to these questions are documented. NMQ can be administered either by a conversation method or can be self-administered [12].

Numerous articles on WMSDs have been published previously, and there seems to be a steady increase in the number of publications on the subject. A bibliometric review on RULA by Gómez-Galan et al. [15] only reported a country-wise, year-wise, and journal-wise analysis, while citations and other analyses were not reported. Despite their importance, the most influential research articles on the use of RULA and NMQ in healthcare practitioners have not been previously reported. The total number of citations of a research article is an indicator of the influence wielded in the area of interest [16]. A citation breakdown is a bibliometric analysis to calculate the comparative reputation of a scientific research paper by investigating the citations credited to that research paper [17]. In recent years, the highest-cited research articles have been detected by utilizing citation scrutiny in numerous medical fields [18,19,20,21,22]. 

Assessing bibliometrics is the arena of the medical field, which utilizes approaches such as citation analysis to estimate the implementation of research [23]. The total number of citations of formerly published work in scientific papers indicates its following identification in a particular area of study. It has an additional effect on the scientific community [24]. Bibliometric studies are mainly valuable for guiding healthcare specialists in creating evidence-based judgments in clinical practice. Along with this, the topmost cited papers in research are generally considered the most influential research within a particular field, raising innovative research ideas [25]. Therefore, a detailed bibliometric analysis of the topmost cited research articles can benefit the understanding of a specific discipline, which aids in generating useful guidelines in the field of biomedical research [26].

Therefore, we conducted analyses on the citations, subject-wise contribution, quartile-wise contribution, primary author contribution, institutional-wise, study type, and evidence level of the top 50 most-cited articles regarding the RULA and NMQ.

## 2. Methods

### 2.1. Collection of Articles

We searched for all the relevant articles on the RULA and NMQ using Web of Science and MEDLINE records to identify the top 50 most commonly cited research articles in various healthcare-related specialties [27,28,29]. The following search terms were used: Ergonomics, assessment, workplace, musculoskeletal disorders, RULA, and NMQ. The search was executed on 20 June 2021 and yielded 429 results related to the RULA and 710 concerning the NMQ. 

### 2.2. Inclusion Criteria 

Only original articles reporting the use of the RULA and/or NMQ in healthcare professionals were included in this bibliometric analysis. 

### 2.3. Exclusion Criteria

Original articles reporting the use of the RULA and/or NMQ in non-healthcare practitioners, narrative reviews, systematic reviews, meta-analyses, letters to the editor, short-communication-type articles, and abstracts of scientific meetings were omitted from the bibliometric analysis [30]. 

### 2.4. Organization of Articles

In total, 1139 studies were identified during the initial screening. Out of these, 376 studies were excluded for not having reported musculoskeletal outcomes. After a careful screening of the remaining 763 studies based on the inclusion standards, 544 studies were excluded due to inadequate data, failure to address the research question, dealing with non-healthcare practitioners, or duplicates, and the remaining 219 studies met the inclusion criteria. These articles were organized according to the number of citations, and the top 50 most-cited articles regarding the RULA and NMQ were incorporated into the final analysis (Figure 1).

### 2.5. Extraction of Articles

Two autonomous assessors selected suitable studies based on the inclusion and exclusion criteria. Then, the reviewers identified the full texts of all relevant articles and selected the top 50 cited publications regarding the RULA and NMQ. 

The following data were extracted from the included articles: Title of the paper, primary author’s name, journal name, time of publication, impact feature of the journal in 2020, the total number of citations of the research paper, study design, article type, average citations per year, mentions, country, organization of basis, authorship, journal, and level of evidence. Citation statistics for all the selected research articles were achieved by exploring the SCOPUS records. The level of evidence of the articles was determined based on the Centre for Evidence-Based Medicine (CEBM) guidelines, Oxford, UK.

### 2.6. Statistical Analysis

Two reviewers individually agreed on the level of evidence for the research articles. The agreement was outstanding for the level of evidence, with an intraclass correlation coefficient of 0.90 [30]. Two assessors decided on all variances through conversation, and another assessor was involved when no agreement was reached. Data analysis was performed using SPSS software (version 24.0 for Windows; SPSS, Inc., Chicago, IL, USA).

## 3. Results

A total of 1139 articles were found from the search, established on the citations, then the 50 most prominent articles concerning the RULA and NMQ regarding WMSDs were selected for the study.

### 3.1. Citation Breakdown

The 50 most commonly cited articles using the RULA and NMQ in regard to WMSDs have 2011 citations overall (Table 1). These range from 1 to 50 concerning the RULA with a mean of 11.36 (SD = 11.32) mentions per article, with a total of 552 citations, as shown in Table 1. Regarding NMQ, the number of citations ranged from 13 to 106, with a mean of 30.66 (SD = 18.30) for each article, with a total of 1518 citations, as shown in Table 1. 

Concerning the RULA, the research article with the highest citations was a paper by Lawson et al. in 2007 [31]. Regarding the NMQ, the research article with the highest citations was a paper by Smith et al., which was published in 2006 [32].

Moreover, the greater length of an article reflects the increased scientific complexity and higher methodological quality of a study; in addition, lengthier articles are expected to contain more information, thus increasing the possibility that part of it will be appropriate to be cited by other researchers.

### 3.2. Journal Analysis

The top 50 cited original articles on RULA were published in 35 journals. Work—A Journal of Prevention, Assessment & Rehabilitation (six articles) was the most commonly cited journal followed by Surgical Endoscopy and Other Interventional Techniques and the Journal of Robotic surgery (each with three articles). Furthermore, the International Nursing Review, Otolaryngology—Head and Neck Surgery, Applied Ergonomics, the Indian Journal of Dental Research, the Journal of Physical Therapy Science, and the Journal of Back Musculoskeletal Rehabilitation each had two articles. Another 26 journals contributed a single article each for this analysis (Table 2).

Concerning the NMQ, 39 diverse journals were associated with the 50 most-cited scientific articles in assessing the WMSDs related to healthcare practitioners. Work—A Journal of Prevention, Assessment & Rehabilitation (four articles) ranked first in the total contribution followed by Industrial Health, BMC Research Notes, Applied Ergonomics, the International Journal of Dental Hygiene, BMC Musculoskeletal Disorders, Revista Latino Americana de Enfermagem, the American Journal of Industrial Medicine, and International Archives of Occupational and Environmental Health (each with two articles). Another 30 journals contributed a single article each for this analysis (Table 3).

### 3.3. Subject Wise

Regarding the RULA, according to a subject-wise analysis, the majority of the research papers were published under the headings of Musculoskeletal disorders (*n* = 18; citations: 179), Work (*n* = 13; citations: 102), and Ergonomics (*n* = 8; citation: 110), followed by Risk factors (*n* = 5; citations: 45) (Table 4).

In regard to the NMQ, according to the subject-wise analysis, the majority of the research papers were published under the heading of Musculoskeletal disorders (*n* = 41; citations: 1122); Work (*n* = 16; citations: 439), and Risk factors (*n* = 10; citation: 106), followed by Ergonomics (*n* = 2; citations: 37) (Table 4).

### 3.4. Quartile Wise

In regard to RULA, among all 35 journals, 10 journals (28.5%) were ranked in the first quartile (*n* = 225 citations), 1 (2.8%) in the second (*n* = 9 citations), 9 (25.7%) in third (*n* = 57 citations), 7 (20%) in the fourth (*n* = 112 citations), and 8 (22.8%) in the non-quartile category (*n* = 149 citations) (Table 5).

In regard to NMQ, among all 40 journals, 7 journals (17.5%) were ranked in the first quartile (*n* = 233 citations), 7 (17.5%) in the second (*n* =335 citations), 12 (30%) in third (*n* = 430 citations), 5 (12.5%) in the fourth (*n* = 267 citations), and 9 (22.5%) (*n* = 253 citations) in the non-quartile category (Table 5).

The significance of dividing the journals into quartiles indicates where a journal’s standing lies in a specific subject classification. These quartiles rank the journals from highest to lowest, established on their impact factor or impact index. A quartile is a category of scientific journals that shows their credibility, reliability, and quality. The quartile also reflects the demand for the journal by the scientific community.

### 3.5. Oldest and Latest Articles

All 50 highly cited articles on WMSDs concerning the RULA and NMQ were published in English over the past 28 years, from 1993 to 2021. The oldest article in relation to the RULA was published in the British Journal of Biomedical Science by Kilroy et al. (2000). The oldest article in relation to the NMQ was published in the Journal of Dental Hygiene by Oberg et al. (1993).

It was shown that the length of time and paper citation count also depend on the length of the article, number of references, title length, type of affiliations, and number of authors. However, all those factors are independent of the scientific merit of publication.

Year-wise distribution: The highest output of the top-cited articles in relation to the RULA was noted in the 2010s wherein 33 articles of the top 50 most-cited articles were published (Figure 2a). In regard to the NMQ, the highest output of the top cited articles was noted in the 2010s wherein 37 articles of the top 50 were published (Figure 2b).

### 3.6. Authors

Five authors published more than one research article related to the RULA (Table 6). Two authors published more than one research article related to the NMQ (Table 6).

### 3.7. Countries

The top-cited articles in WMSDs in healthcare practitioners in regard to the RULA originated from 13 countries. The country delivering the most articles was the USA (*n* = 18), followed by Iran (*n* = 11), India and Brazil (*n* = 4), Sweden (*n* = 3), and Germany and England (*n* = 2). The remaining articles were from other respective countries. Therefore, the geographic distribution of these most-cited articles is represented in Figure 3a.

The top-cited articles in WMSDs in healthcare practitioners in regard to the NMQ originated from 20 countries. The countries delivering the highest number of articles were Iran (*n* = 3), followed by Australia (*n* = 7), the USA (*n* = 5), France, Portugal, and Brazil (*n* = 3). The remaining articles were from the other respective countries. Therefore, the geographic distribution of these most-cited articles is represented in Figure 3b.

### 3.8. Institutions

A total of 50 institutions produced the top-cited articles in WMSDs concerning the RULA. Three institutions published three or more of the most commonly cited articles. Thirteen (26%) of these institutions were located in the USA, followed by 8 (16%) in Iran, 4 (8%) in Brazil and India, 3 (6%) in Sweden, and 2 (4%) in Germany, Canada, Turkey, Canada, South Korea, and England. The remaining countries contributed one article each. The institutions producing the most articles were Ohio State University, Tehran University, and the University of Pittsburgh (*n* = 3), followed by the University of North Carolina and Iran University of Medical Sciences (*n* = 2), and the remaining Universities contributed one article each (Table 7).

A total of 49 institutions were involved in producing the top-cited articles in WMSDS concerning the NMQ. Three institutions published three or more of the most commonly cited articles. Seven (14.2%) of these institutions were located in Iran, followed by six (12.2%) in Australia, five (10.2%) in the USA, three (6.1%) in Sweden, China, and Iran, and two (4%) in France, Saudi Arabia, and Egypt. The remaining countries contributed one article each. The institutions producing the most articles were Tehran University, Shiraz University, and Babol University (*n* = 3), followed by Universidade Nova de Lisboa, Queensland University, National Institute of Industrial Health, and King Abdulaziz University (*n* = 2), and the remaining universities contributed one article each (Table 7).

Ranking institutions by country may help to build global brand visibility, forge strategic partnerships, and recruit international talent. It also allows for analyses according to academic reputation, employer reputation, research citations per paper, h-index, and the International Research Network.

### 3.9. Study Designs of Articles

The main stream of the top 50 cited research articles associated with both the RULA and the NMQ were cross-sectional studies, followed by observational, comparative, experimental, and other types of studies [33] (Table 8). 

### 3.10. Level of Evidence

The top 50 most-cited articles could be classified into all evidence levels (Table 9). The majority of the articles were within evidence level III (*n* = 70), followed by evidence level IV (*n* = 23), evidence level 1B (*n* = 4), and evidence level IIB (*n* = 3) [34].

## 4. Discussion

To the best of our knowledge, this is the first bibliometric analysis of papers on work-related musculoskeletal disorders concerning the RULA and NMQ. The total number of citations for each paper, the unique bibliometric indicator, is a valuable tool to determine the impact of publications. From the analysis of these top 50 most-cited articles published, we attempted to determine what fundamentals of a research article make it vastly effective. The top-cited article in our list in relation to the RULA had 50 citations [31], and that related to the NMQ had 106 citations [32]. This number is much lower than the WMSDs in relation to non-healthcare specialties [35,36]. The citations varied between specialties, predominantly reliant on the number of investigators in exact therapeutic fields [4,37,38]. Identifying those classic articles is beneficial for better insight into the history and progress of evaluation of WMSDs by using RULA and NMQ among healthcare practitioners and also for planning future research.

In relation to our present study, open access may be another significant factor in drawing citations [39,40]. It is widely known that open-access scientific papers are more easily accessible and cited compared to non-open scientific papers [41,42]. However, we also found that articles concerning otolaryngologists and neurosurgeons in open-access format did not obtain considerably more citations than non-open-access articles in the same specialties [38,43]. It can thereby be inferred that the extent to which open-access articles increase citations varies depending on the specialty. 

All top 50 cited articles in WMSDs in relation to the RULA and NMQ were published in English. This establishes that English is the most commonly used scholarly language in WMSDs. Additionally, all of the most-cited papers occurred in 35 journals in relation to RULA and 40 in relation to NMQ. In relation to individual contributions of the journal published, the most productive journal was Work—A Journal of Prevention, Assessment & Rehabilitation, with ten articles (both RULA and NMQ) in the top list. This result suggests that this journal was the most preferred in regard to WMSDs.

In relation to quartile classifications, these were analyzed for individual journals in each theme classification, corresponding to which quartile of the scientific journal dominates in the impact factor division of that subject category [44]. The total number of publications and/or the distribution of total publications in a particular quartile generally corresponded to the first quartile (Q1) in regard to the RULA and the third quartile (Q3) in regard to the NMQ. Moreover, in the present study, the overall greater number of cited articles occurred in the third quartile (Q3).

With relation to authors and institutions, good publication track records in WMSDs related to healthcare practitioners have been determined in the present study [45,46,47,48,49]. Tehran University has great priority in this field by contributing six publications overall. Musculoskeletal disorders are the most prevalent theme, with approximately 18 articles related to the RULA and 41 related to the NMQ, which emphasizes the significance of musculoskeletal disorders in subspecialties such as joints, bones, muscles, etc.

In regard to study type, we demonstrated that cross-sectional studies were the most popular study type in this study. This finding is consistent with many studies assessing WMSDs [50,51,52,53]. Furthermore, most of the top cited articles were level III evidence. This result indicates that the level of evidence is certainly not a positive aspect of the total number of citations [54]. This is likely because original ideas and assessments are initially published as cross-sectional studies and still catch the attention of practitioners or researchers.

In relation to the sequential allocation of citations, the 2001–2010 time period had the highest number of mentions on our 50 top-cited research articles list related to WMSDs in healthcare practitioners; 33 articles were related to the RULA, and 37 were related to the NMQ. There are various reasons for this result in the present study. First, it could take approximately 10 years or further for prominent article citations to reach their peak, as acknowledged through bibliometric examination [31,32]. Second, researchers tend to underestimate the influence of the most recent studies compared to old studies [55]. Third, some research scholars believe that the accurate importance of the research papers cannot be considered until at least 10–20 years afterward the date of publication in the journal [56]. Therefore, research articles published recently need additional time to gather citations to establish their importance.

Lastly, in relation to the geographical distribution, more than one-third of the top-cited WMSD-related articles were from the United States. It is no wonder that the United States has benefited in the rankings, as various healthcare sectors have found similar results [57,58] due to various aspects: (1) The American research and scientific community is substantially bigger than that of any other community in the world; (2) the availability of more funding for research; and (3) previous research studies have pointed out that there is a predisposition for researchers from the United States to cite local articles [59]. This may result in the authors from the US having an essential advantage when trying to publish the most cited and dominant research articles.

### Limitations of the Study

There are several limitations in the present study. First, we only utilized the Web of Science and MEDLINE for our analysis; consequently, some research papers indexed by Google Scholar or SCOPUS or basic medical journal databases may have been missed. Second, there was an association with the publication phase, with newly published research articles experiencing a disadvantage in relation to the citation calculation. With this issue, possibly more appropriate and prominent research articles published recently may not have been included in the top 50 cited articles list because they have not had sufficient time to gather the required number of citations. Lastly, possible factors that influence the citations cannot be exactly established in this analysis; these include researcher self-mentions, citations in workbooks, meeting notes, lectures, symposiums, and web-based writings [57,60]. Nevertheless, the statistics delivered in the present work do provide insight into the most important investigation areas of WMSDs.

## 5. Conclusions

The top-cited papers were all published in the English language, originated from institutions in the United States, and were primarily cross-sectional studies with level III evidence. Most articles were published in Work—A Journal of Prevention, Assessment & Rehabilitation. This top citation list offers valuable insight into the assessment of WMSDs using the RULA and NMQ, especially in the healthcare sector, and serves as a crucial basis of evidence for investigators and research scholars for future research on WMSDs.

## Figures and Tables

**Figure 1 ijerph-20-01932-f001:**
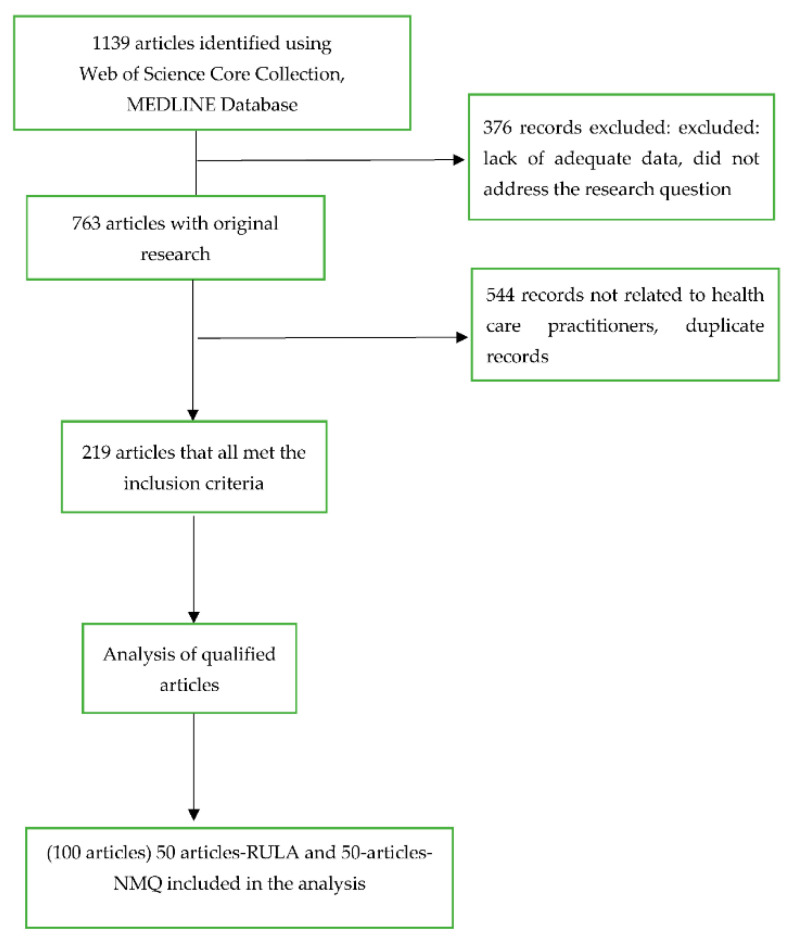
Flowchart illustrating the procedure of allocation of articles.

**Figure 2 ijerph-20-01932-f002:**
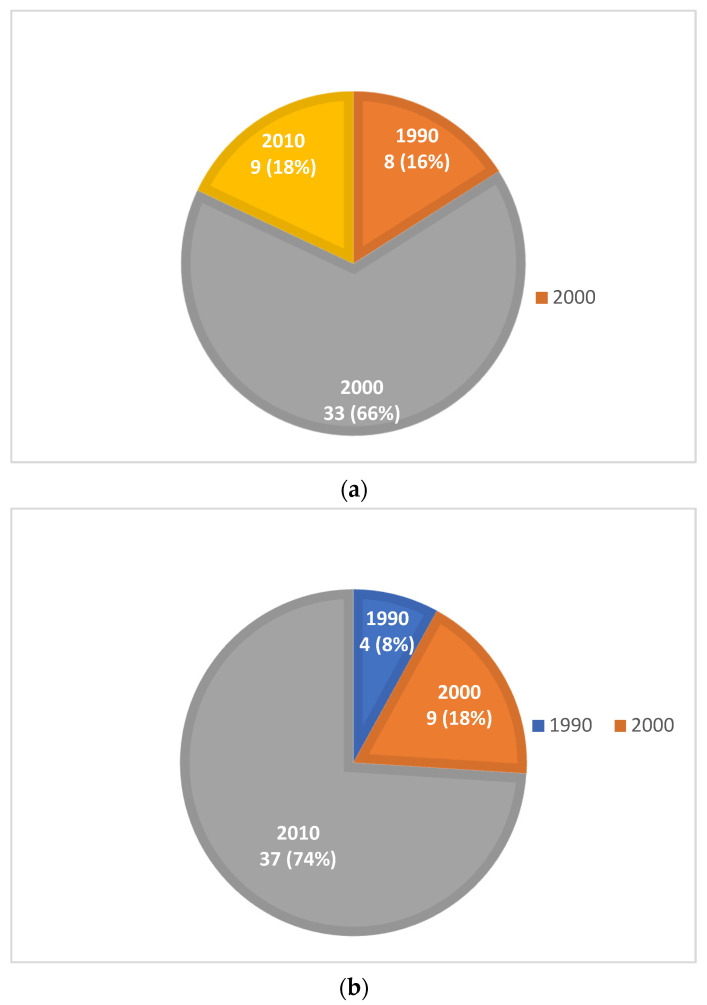
(**a**): Time distribution of the top 50 most-cited articles in WMSDS related to healthcare practitioners using RULA. (**b**): Time distribution of the top 50 most-cited articles in WMSDS related to healthcare practitioners using NMQ.

**Figure 3 ijerph-20-01932-f003:**
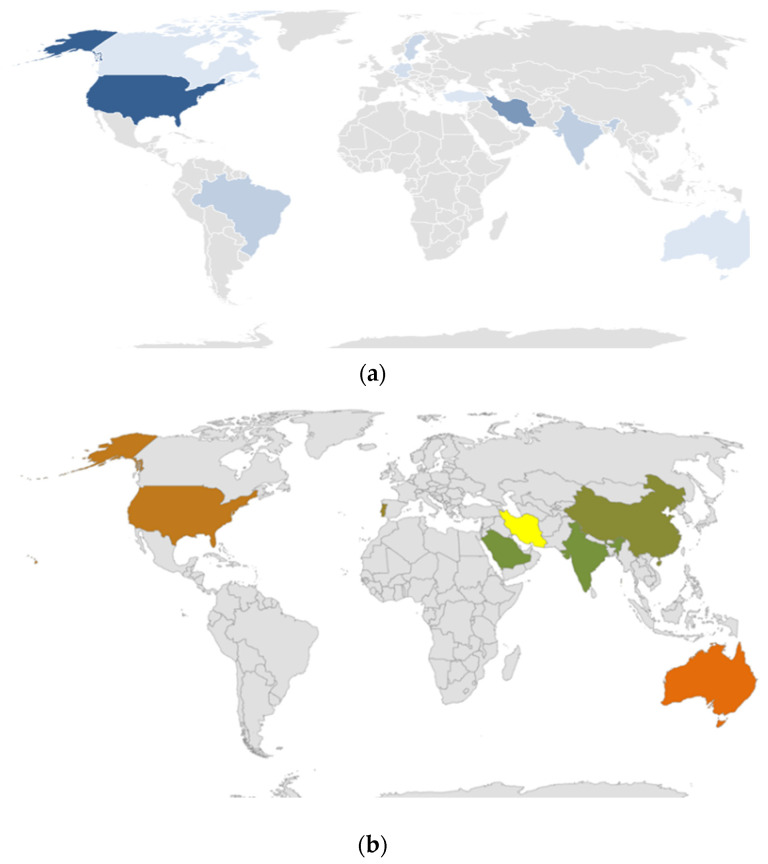
(**a**): Geographic distribution of the top 50 most-cited articles in WMSDS using RULA in regard to healthcare practitioners. (**b**): Geographic distribution of the top 50 most-cited articles in WMSDS using NMQ in regard to healthcare practitioners.

**Table 1 ijerph-20-01932-t001:** (a): List of top 50 cited articles in WMSDS using RULA concerning healthcare practitioners. (b): List of top 50 cited articles in WMSDS using NMQ in regard to healthcare practitioners.

Title & Year	Journal	Total Citations
(**a**)		
Postural ergonomics during robotic and laparoscopic gastric bypass surgery: a pilot project. (2007)	*J. Robot Surg.*	50
Ergonomics and human factors in endoscopic surgery: a comparison of manual vs. telerobotic simulation systems. (2005)	*Surg. Endosc.*	40
Ergonomic deficits in robotic gynecologic oncology surgery: a need for intervention. (2013)	*J. Minim. Invasive Gynecol.*	36
Ergonomic analysis of microlaryngoscopy. (2010)	*Laryngoscope*	27
Two posture analysis approaches and their application in a modified rapid upper limb assessment evaluation. (2007)	*Ergonomics*	25
Posture stress on firefighters and emergency medical technicians (EMTs) associated with repetitive reaching, bending, lifting, and pulling tasks. (2010)	*Work*	24
Laparoscopic cholecystectomy poses physical injury risk to surgeons: analysis of hand technique and standing position. (2011)	*Surg. Endosc.*	24
Automated high-frequency posture sampling for ergonomic assessment of laparoscopic surgery. (2001)	*Surg. Endosc.*	22
A study of computer-related upper limb discomfort and computer vision syndrome. (2007)	*J. Hum. Ergol. (Tokyo)*	22
Assessment of dental student posture in two seating conditions using RULA methodology—a pilot study. (2007)	*Br. Dent. J.*	22
Predisposing factors for musculoskeletal symptoms in intensive care unit nurses. (2015)	*Int. Nurs. Rev.*	19
Postural Assessment of Students Evaluating the Need of Ergonomic Seat and Magnification in Dentistry. (2014)	*J. Indian Prosthodont. Soc.*	19
Predisposing factors for musculoskeletal symptoms in intensive care unit nurses. (2015)	*Int. Nurs. Rev.*	19
Musculoskeletal disorders among a group of Iranian general dental practitioners. (2015)	*J. Back Musculoskelet Rehabil.*	18
Effect of chair types on work-related musculoskeletal discomfort during vaginal surgery. (2016)	*Am. J. Obstet. Gynecol.*	16
Ergonomic intervention: its effect on working posture and musculoskeletal symptoms in female biomedical scientists. (2000)	*Br. J. Biomed. Sci.*	15
The association between restorative pre-clinical activities and musculoskeletal disorders. (2014)	*Eur. J. Dent. Educ.*	15
Analysis of the risk factors of musculoskeletal disease among dentists induced by work posture. (2015)	*J. Phys. Ther. Sci.*	15
Do dental students have a neutral working posture? (2016)	*J. Back Musculoskelet Rehabil.*	14
Ergonomic risk factors and their association with musculoskeletal disorders among Indian dentist: a preliminary study using Rapid Upper Limb Assessment. (2014)	*Indian J. Dent. Res.*	12
Approach of industrial physical therapy to assessment of the musculoskeletal system and ergonomic risk factors of the dental hygienist. (2013)	*J. Phys. Ther. Sci.*	12
Contribution of positioning to work-related musculoskeletal discomfort in diagnostic medical sonographers. (2014)	*Work*	10
Optimizing Positioning for In-Office Otology Procedures. (2017)	*Otolaryngol. Head. Neck. Surg.*	8
Impact of a simulation-based ergonomics training curriculum on work-related musculoskeletal injury risk in colonoscopy. (2020)	*Gastrointest. Endosc.*	8
Working postures of dental students: ergonomic analysis using the Ovako Working Analysis System and rapid upper limb assessment. (2013)	*Med. Lav.*	7
Comparing the Effectiveness of Three Ergonomic Risk Assessment Methods-RULA, LUBA, and NERPA-to Predict the Upper Extremity Musculoskeletal Disorders. (2018)	*Indian J. Occup. Environ. Med.*	6
Ergonomic Risk Factors and Their Association With Lower Back and Neck Pain Among Pharmaceutical Employees in Iran. (2016)	*Workplace Health Saf.*	6
Surgeons’ muscle load during robotic-assisted laparoscopy performed with a regular office chair and the preferred of two ergonomic chairs: A pilot study. (2019)	*Appl. Ergon.*	6
Differences in dentists’ working postures when adopting proprioceptive derivation vs. conventional concept. (2005)	*Int. J. Occup. Saf. Ergon.*	5
Assessment of upper limb musculoskeletal pain and posture in workers of packaging units of pharmaceutical industries. (2017)	*Work*	5
Patient Positioning During In-Office Otologic Procedures Impacts Physician Ergonomics. (2018)	*Otol. Neurotol.*	4
Objective ergonomic risk assessment of wrist and spine with motion analysis technique during simulated laparoscopic cholecystectomy in experienced and novice surgeons. (2017)	*J. Minim. Access Surg.*	4
Analysis of comfort and ergonomics for clinical work environments. (2016)	*Annu. Int. Conf. IEEE Eng. Med. Biol. Soc.*	4
Hand motion analysis for assessment of nursing competence in ultrasound-guided peripheral intravenous catheter placement. (2019)	*J. Vasc. Access*	3
Design and ergonomic assessment of an infusion set connector tool used in nursing work. (2019)	*Appl. Ergon.*	3
Study of musculoskeletal risks of the office-based surgeries. (2012)	*Work*	3
Risk of musculoskeletal disorders in upper limbs in dental students: concordance of different methods for estimation of body angle. (2013)	*Indian J. Dent. Res.*	3
Reliability, Construct Validity and Interpretability of the Brazilian version of the Rapid Upper Limb Assessment (RULA) and Strain Index (SI). (2018)	*Braz. J. Phys. Ther.*	3
Estimation of surgeons’ ergonomic dynamics with a structured light system during endoscopic surgery. (2019)	*Int. Forum. Allergy Rhinol.*	3
Use of the Omaha System to identify musculoskeletal problems in intensive care unit nurses: a case study. (2019)	*Br. J. Nurs.*	2
Ergonomic assessment of robotic general surgeons: a pilot study. (2020)	*J. Robot Surg.*	2
Evaluating the efficacy of an educational ergonomics module for improving slit lamp positioning in ophthalmology residents. (2019)	*Can. J. Ophthalmol.*	2
Ergonomic assessment of the first assistant during robot-assisted surgery. (2019)	*J. Robot Surg.*	2
Risk factors for musculoskeletal disorders in an obstetrician-gynecologist and orthopedic surgeon. (2020)	*Work*	1
Impact of Workspace Design on Radiation Therapist Technicians’ Physical Stressors, Mental Workload, Situation Awareness, and Performance. (2021)	*Pract. Radiat. Oncol.*	1
A 10-week exercise intervention can improve work posture but not neck/shoulder symptoms in dental health students: A pilot cohort study. (2020)	*Work*	1
An intuitive surgical handle design for robotic neurosurgery. (2021)	*Int. J. Comput. Assist. Radiol. Surg.*	1
Relationships between the postures of dentists and chairside dental assistants. (2020)	*J. Dent. Educ.*	1
Quantitative Assessment of Surgical Ergonomics in Otolaryngology. (2020)	*Otolaryngol. Head. Neck. Surg.*	1
SOPEZ: study for the optimization of ergonomics in the dental practice—musculoskeletal disorders in dentists and dental assistants: a study protocol. (2020)	*J. Occup. Med. Toxicol.*	1
(**b**)		
A detailed analysis of musculoskeletal disorder risk factors among Japanese nurses. (2006)	*J. Saf. Res.*	106
Musculoskeletal disorders among female dental personnel–clinical examination and a 5-year follow-up study of symptoms. (1999)	*Int. Arch. Occup. Environ. Health*	82
Musculoskeletal complaints and psychosocial risk factors among Chinese hospital nurses. (2004)	*Occup. Med. (Lond.)*	73
Musculoskeletal symptoms among dentists in relation to work posture. (2000)	*Work*	59
Development and test-retest reliability of an extended version of the Nordic Musculoskeletal Questionnaire (NMQ-E): a screening instrument for musculoskeletal pain. (2009)	*J. Pain.*	58
Perceived demands and musculoskeletal disorders in operating room nurses of Shiraz city hospitals. (2010)	*Ind. Health*	52
The influence of perceived stress and musculoskeletal pain on work performance and work ability in Swedish health care workers. (2014)	*Int. Arch. Occup. Environ. Health*	44
Association between perceived demands and musculoskeletal disorders among hospital nurses of Shiraz University of Medical Sciences: a questionnaire survey. (2006)	*Int. J. Occup. Saf. Ergon.*	43
Association between psychosocial factors and musculoskeletal symptoms among Iranian nurses. (2010)	*Am. J. Ind. Med.*	39
Job stress dimensions and their relationship to musculoskeletal disorders in Iranian nurses. (2014)	*Work*	39
Musculoskeletal problems among Ontario dental hygienists. (1995)	*Am. J. Ind. Med.*	35
Work related musculoskeletal disorders in primary health care nurses. (2017)	*Appl. Nurs. Res.*	34
[Self-reported musculoskeletal symptoms among nursing personnel]. (2003)	*Rev. Lat. Am. Enfermagem.*	33
Prevalence of multisite musculoskeletal symptoms: a French cross-sectional working population-based study. (2012)	*BMC Musculoskelet Disord.*	33
Prevalence and risk factors associated with musculoskeletal discomfort in New Zealand veterinarians. (2010)	*Appl. Ergon.*	32
Frequency and risk factors of musculoskeletal pain in nurses at a tertiary centre in Jeddah, Saudi Arabia: a cross sectional study. (2014)	*BMC Res. Notes*	32
Prevalence of Upper Extremity Musculoskeletal Disorders in Dentists: Symptoms and Risk Factors. (2015)	*J. Environ. Public Health*	32
Psychosocial aspects of work and musculoskeletal disorders in nursing workers. (2010)	*Rev. Lat. Am. Enfermagem*	31
Work-related injuries among physiotherapists in public hospitals: a Southeast Asian picture. (2011)	*Clinics (Sao Paulo)*	30
Musculoskeletal complaints in dental hygiene: a survey study from a Swedish county. (1993)	*J. Dent. Hyg.*	30
Relationship between musculoskeletal disorders, job demands, and burnout among emergency nurses. (2012)	*Adv. Emerg. Nurs. J.*	29
Psychosocial stress and multi-site musculoskeletal pain: a cross-sectional survey of patient care workers. (2013)	*Workplace Health Saf.*	29
Prevalence and risk factors for foot and ankle musculoskeletal disorders experienced by nurses. (2014)	*BMC Musculoskelet Disord.*	29
Musculoskeletal disorder risk factors among nursing professionals in low resource settings: a cross-sectional study in Uganda. (2014)	*BMC Nurs.*	28
Predictors of work-related musculoskeletal disorders among dental hygienists. (2012)	*Int. J. Dent. Hyg.*	27
Prevalence and factors associated with neck, shoulder and low back pains among medical students in a Malaysian Medical College. (2013)	*BMC Res. Notes*	24
Musculoskeletal pain in resident orthopaedic surgeons: results of a novel survey. (2014)	*Iowa Orthop. J.*	24
Self reported symptoms in the neck and upper limbs in nurses. (1996)	*Appl. Ergon.*	24
A study on job postures and musculoskeletal illnesses in dentists. (2013)	*Int. J. Occup. Med. Environ. Health*	23
Association between shift working and musculoskeletal symptoms among nursing personnel. (2014)	*Iran J. Nurs. Midwifery Res.*	23
Hospital nurses tasks and work-related musculoskeletal disorders symptoms: A detailed analysis. (2015)	*Work*	23
Occupational risk factors for upper-limb and neck musculoskeletal disorder among healthcare staff in nursing homes for the elderly in France. (2014)	*Ind. Health*	22
Feasibility and acceptance of a robotic surgery ergonomic training program. (2014)	*JSLS*	22
Work-related musculoskeletal disorders of perioperative personnel in the Netherlands. (2007)	*AORN J.*	21
Prevalence of work-related musculoskeletal complaints among dentists in India: a national cross-sectional survey. (2013)	*Indian J. Dent. Res.*	21
Musculoskeletal complaints among a group of Turkish nurses. (2005)	*Int. J. Neurosci*	21
Prevalence of and risk factors for low back pain among dentists. (2015)	*J. Phys. Ther. Sci.*	20
Prevalence of musculoskeletal symptoms in hospital nurse technicians and licensed practical nurses: associations with demographic factors. (2014)	*Braz. J. Phys. Ther.*	19
Musculoskeletal disorders among a group of Iranian general dental practitioners. (2015)	*J. Back Musculoskelet Rehabil*	18
Musculoskeletal Disorders and Working Posture among Dental and Oral Health Students. (2016)	*Healthcare (Basel)*	18
Links between nurses’ organisational work environment and upper limb musculoskeletal symptoms: independently of effort-reward imbalance! The ORSOSA study. (2011)	*Pain*	18
Musculoskeletal disorders among robotic surgeons: a questionnaire analysis. (2014)	*Arch. Ital. Urol. Androl.*	18
Validity and reliability of an online extended version of the Nordic Musculoskeletal Questionnaire (NMQ-E2) to measure nurses’ fitness. (2015)	*J. Clin. Nurs.*	17
High and specialty-related musculoskeletal disorders afflict dental professionals even since early training years. (2013)	*J. Appl. Oral. Sci.*	16
Low back pain among nurses in Slovenian hospitals: cross-sectional study. (2017)	*Int. Nurs. Rev.*	15
Ergonomic intervention: its effect on working posture and musculoskeletal symptoms in female biomedical scientists. (2000)	*Br. J. Biomed. Sci.*	15
Assessing work-related musculoskeletal symptoms among otolaryngology residents. (2017)	*Am. J. Otolaryngol.*	13
Prevalence of musculoskeletal disorders among dentists in the Hail Region of Saudi Arabia. (2015)	*Ann. Saudi Med.*	13
Effects of patient-handling and individual factors on the prevalence of low back pain among nursing personnel. (2017)	*Work*	13
The Prevalence of and Risk Factors Associated with Musculoskeletal Disorders among Sonographers in Central China: A Cross-Sectional Study. (2016)	*PLoS ONE*	13

**Table 2 ijerph-20-01932-t002:** Top Journals with their individual contribution to the 50 most-cited articles on WMSDs using RULA related to healthcare practitioners.

S.No.	Journal Name	Impact Factor (Clarivate Analytics) (2020)	Quartile	Category	Number of Articles
1	WORK—A Journal of Prevention, Assessment & Rehabilitation	1.50	4	Public, Environmental & Occupational Health	6
2	Surgical Endoscopy and Other Interventional Techniques	4.58	1	Surgery	3
3	Journal of Robotic surgery	n/a	n/a	Surgery	3
4	International Nursing Review	2.8	1	Nursing	2
5	Otolaryngology-Head and Neck Surgery	3.4	1	Otorhinolaryngology, Surgery	2
6	Applied Ergonomics	3.6	2	Ergonomics, Industrial, Psychological, Applied	2
7	Journal of Back and Musculoskeletal Rehabilitation	1.3	4	Orthopaedics, Rehabilitation	2
8	Indian Journal of Dental Research	n/a	n/a	Dentistry	2
9	Journal of Physical Therapy Science	n/a	n/a	Rehabilitation	2
10	Practical Radiation Oncology	3.5	3	Oncology, Radiology, Nuclear medicine & Medical Imaging Sciences	1
11	Journal of Dental Education	2.2	3	Dentistry, Oral Surgery & Medicine	1
12	International Journal of Computer Assisted Radiology and Surgery	2.9	3	Engineering, Biomedical, Radiology, Nuclear medicine & Medical imaging	1
13	Journal of Occupational Medicine and Toxicology	2.6	3	Public, Environmental & Occupational Health	1
14	British Journal of Nursing	n.a	n.a	Nursing	1
15	British Journal of Biomedical science	3.8	1	Medical laboratory technology	1
16	European Journal of Dental Education	2.3	3	Dentistry, Oral Surgery & Medicine, Education, Scientific disciplines	1
17	American Journal of Obstetrics and Gynecology	8.6	1	Obstetrics and Gynecology	1
18	Journal of Human Ergology (Tokyo)	n/a	n/a	Ergology	1
19	British Dental Journal	1.6	4	Dentistry, Oral Surgery & Medicine	1
20	Ergonomics	2.7	3	Engineering, Industrial, Psychology	1
21	Laryngoscope	3.3	1	Otorhinolaryngology, Medicine, Research & Experimental	1
22	Journal of Minimally Invasive Gynecology	4.1	1	Obstetrics & Gynecology	1
23	Journal of Indian Prosthodontic Society	n.a	n.a	Dentistry, Oral Surgery & Medicine	1
24	Canadian Journal of Ophthalmology	1.8	4	Ophthalmology	1
25	International Forum of Allergy & Rhinology	3.8	1	Otorhinolaryngology	1
26	Otology & Neurotology	2.3	3	Otorhinolaryngology, Clinical Neurology	1
27	Journal of Minimal Access Surgery	1.4	4	Surgery	1
28	Annual International Conference of the IEEE Engineering in Medicine and Biology Society	n/a	n/a	-	1
29	International Journal of Occupational Safety and Ergonomics	2.1	3	Public, Environmental & Occupational Health, Poland	1
30	Indian Journal of Occupational and Environmental Medicine	n/a	n/a	Public, Environmental & Occupational Health, India	1
31	Workplace Health & Safety	1.4	3	Nursing, USA	1
32	Brazilian Journal of Physical Therapy	3.3	1	Rehabilitation, Orthopedics	1
33	Journal of Vascular Access	2.2	4	Peripheral Vascular Disease	1
34	Medicina del Lavoro	1.2	4	Public, Environmental & Occupational Health, Italy	1
35	Gastrointestinal Endoscopy	9.4	1	Gastroenterology & Hepatology, USA	1

**Table 3 ijerph-20-01932-t003:** Top Journals with their contribution to the 50 most-cited articles on WMSDs using NMQ related to healthcare practitioners.

S.No.	Journal Name	Impact Factor (Clarivate Analytics) (2020)	Quartile	Category	Number of Articles
1	WORK—A Journal of Prevention, Assessment & Rehabilitation	1.50	4	Public, Environmental & Occupational Health	4
2	Industrial Health	2.1	3	Public, Environmental & Occupational Health, Environmental Sciences	2
3	BMC Research Notes	n/a	n/a	Research	2
4	Applied Ergonomics	3.6	2	Ergonomics, Industrial, Psychological, Applied	2
5	International Journal of Dental Hygiene	2.4	3	Dentistry, Oral Surgery & Medicine	1
6	BMC Musculoskeletal Disorders	2.3	2	Orthopaedics, Rheumatology	2
7	Revista Latino-Americana de Enfermagem	1.4	3	Nursing	2
8	American Journal Of Industrial Medicine	2.2	3	Public, Environmental & Occupational Health	2
9	International Archives of Occupational and Environmental Health	3.0	2	Public, Environmental & Occupational Health	2
10	Journal Of Pain	5.8	1	Clinical Neurology, Neurosciences	1
11	Occupational Medicine	1.6	4	Public, Environmental & Occupational Health	1
12	Journal Of Safety Research	3.4	1	Public, Environmental & Occupational Health, Transportation	1
13	American Journal Of Otolaryngology	1.8	3	Otorhinolaryngology	1
14	PLoS ONE	3.2	2	Multidisciplinary Sciences	1
15	International Nursing Review	2.8	1	Nursing	1
16	British Journal of Biomedical science	3.8	1	Medical laboratory technology	1
17	Journal of Applied Oral Science	2.6	2	Dentistry, Oral Surgery & Medicine	1
18	Journal Of Clinical Nursing	3.0	1	Nursing	1
19	Journal of Back and Musculoskeletal Rehabilitation	1.3	4	Orthopaedics, Rehabilitation	1
20	Healthcare	2.6	3	Health Care Sciences & Services, Health Policy & Services	1
21	Pain	6.9	1	Anaesthesiology, Neurosciences	1
22	Archivio Italiano di Urologia e Andrologia	n.a	n.a	Urology & Nephrology	1
23	Brazilian Journal of Physical Therapy	3.3	1	Rehabilitation, Orthopedics	1
24	Journal of Physical Therapy Science	n/a	n/a	Rehabilitation	1
25	AORN Journal	0.6	4	Nursing	1
26	Indian Journal of Dental Research	n/a	n/a	Dentistry	1
27	International Journal Of Neuroscience	2.2	4	Neurosciences	1
28	Annals Of Saudi Medicine	1.5	3	Medicine, General & Internal	1
29	International Journal of Occupational Safety and Ergonomics	2.1	3	Public, Environmental & Occupational Health, Poland	1
30	Workplace Health & Safety	1.4	3	Nursing, USA	1
31	Journal of Environmental and Public Health	n/a	n/a	Public, Environmental & Occupational Health	1
32	Applied Nursing Research	2.2	2	Nursing	1
33	Clinics	2.3	3	Medicine, General & Internal	1
34	BMC Nursing	2.2	2	Nursing	1
35	Advanced Emergency Nursing Journal	n/a	n/a	Nursing	1
36	The Lowa Orthopaedic Journal	n/a	n/a	Orthopaedics	1
37	Iranian Journal of Nursing and Midwifery Research	n/a	n/a	Nursing	1
38	JSLS-Journal of the Society of Laparoendoscopic Surgeons	2.1	3	Surgery	1
39	International Journal of Occupational Medicine and Environmental Health	1.8	3	Public, Environmental & Occupational Health	1
40	Journal of Dental Hygiene	n/a	n/a	Dentistry	1

**Table 4 ijerph-20-01932-t004:** (a) Number and citations of research articles as per subject-wise analysis concerning RULA, (b) number and citations of research articles as per subject-wise analysis in regard to NMQ.

Subject Heading	No. of Articles	Total No. of Citations
(**a**)		
Musculoskeletal disorders	18	179
Work	13	102
Ergonomics	8	110
Risk factors	5	45
(**b**)		
Musculoskeletal disorders	41	1122
Work	16	439
Risk factors	10	106
Ergonomics	2	37

**Table 5 ijerph-20-01932-t005:** (a): Number and citations of research articles as per quartile in regard to RULA. (b): Number and citations of research articles as per quartile in regard to NMQ.

Quartile	No. of Journals	Total No. of Citations (*n*) and Percentage (%)
(**a**)		
1	10	225 (40.7%)
2	1	9 (1.6%)
3	9	57 (10.3%)
4	7	112 (20.2%)
Non-quartile	8	149 (26.9%)
(**b**)		
1	7	233 (15.3%)
2	7	335 (22%)
3	12	430 (28.3%)
4	5	267 (17.5%)
Non-quartile	9	253 (16.6%)

**Table 6 ijerph-20-01932-t006:** (a): First authors of top-cited articles on WMSDs in relation to RULA. (b): First authors of top-cited articles on WMSDs in relation to NMQ.

Authors	Number of Articles
(**a**)	
Nandini Govil	2
William M DeMayo	2
Barry E Hirsch	2
Andrew A McCall	2
D Sezgin	2
M N Esin	2
(**b**)	
Alireza Choobineh	2
Derek R Smith	2

**Table 7 ijerph-20-01932-t007:** (a): Top Institutional distribution of all top 50 cited articles in WMSDS related to healthcare practitioners concerning RULA. (b): Top Institutional distribution of all top 50 cited articles in WMSDS related to healthcare practitioners concerning NMQ.

Institution/Organization	Number
(**a**)	
Ohio State University	3
University of Pittsburgh	3
Tehran University	3
University of North Carolina	2
Iran University of Medical Sciences	2
University of Indianapolis	1
The University of Queensland	1
University College London	1
Twente University	1
Goethe-University	1
National University of Ireland Galway	1
Queen’s University	1
University of Colorado Denver	1
University of Wisconsin	1
Araraquara School of Dentistry	1
Universidade Cidade de São Paulo	1
Universidad de Cantabria	1
D. Y. Patil University	1
Luleå University of Technology	1
University of Southern Denmark	1
University of Toronto	1
University of Southern California	1
IP Dental College	1
Kangwon National University	1
Mashhad University of Medical Sciences	1
Univ Estadual Paulista	1
Daejeon University	1
Babol University	1
Bezmialem Vakif University	1
SMBT Dental College	1
Bezmialem Vakif University	1
University of British Columbia	1
Universidade Federal de São Carlos	1
University of Birmingham	1
University of Maryland	1
University of Central Florida	1
Harvard School of Public Health	1
Stanford University	1
Linköping University	1
University Mainz	1
Istanbul University	1
Shiraz University	1
University of California Davis	1
Hormozgan University	1
Shahid Sadoughi University	1
Karolinska University	1
Zanjan University	1
The Catholic University of Korea	1
M. A. Rangoonwala Dental College	1
Weill Cornell Medical College	1
(**b**)	
Tehran University	3
Shiraz University	3
Babol University	3
Universidade Nova de Lisboa	2
Queensland University	2
National Institute of Industrial Health	2
King Abdulaziz University	2
Boston Medical Center	1
University of Hail	1
Tongji Medical College	1
Angela Boškin Faculty of Health Care	1
Tullamore General Hospital	1
Sichuan University	1
Edith Cowan University	1
The University of Melbourne	1
Institut National de la Santé et de la Recherche Médicale (INSERM)	1
San Paolo Hospital	1
Universidade Federal de São Carlos	1
Catharina Hospital	1
Kasturba Medical College	1
Ataturk University	1
University Hospital Saint-Etienne	1
Rutgers University	1
Mashhad University	1
Malaysian Medical College	1
University of Minnesota	1
National Institute of Occupational Health and Safety	1
University of Newcastle	1
Makerere University	1
Zagazig University	1
City University	1
Universiti Kebangsaan	1
University College of Health Sciences	1
Universidade Federal de Santa Maria	1
Massey University	1
Ontario Ministry of Labour	1
Kerman University	1
Universidade Estadual de Campinas	1
LUNAM University	1
Institute of Stress Medicine	1
Tel Aviv University	1
University Hospital	1
Cairo University	1
Shahroud University	1
Islamic Azad University	1
Huazhong University	1
Murdoch University	1
Universidade de Aveiro	1
University of North Carolina	1

**Table 8 ijerph-20-01932-t008:** Study design of top-cited articles of RULA and NMQ in WMSDs in regard to healthcare practitioners.

Study Design		Articles
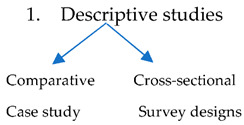	Cross-sectional	44
	Comparative	7
	Case study	4
	Survey design	4
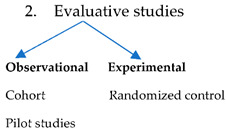	Observational	8
	Cohort	3
	Pilot	3
	Psychometric evaluation	3
	Experimental	5
	Randomized control trial	4
3. Other types		15

**Table 9 ijerph-20-01932-t009:** Level of evidence of the top 50 cited articles in WMSDs in regard to RULA and NMQ.

Level of Evidence Based on CEBM	Type of Article	No of Studies
Level 1B	Randomized Control Trails	4
Level 2B	Cohort studies	3
Level 3	Experimental	5
	Comparative	7
	Psychometric evaluation	3
	Cross sectional	44
	Observational	8
	Pilot	3
Level 4	Survey design	4
	Case study	4
	Other types of research	15

## Data Availability

All relevant data produced or evaluated during this study are included in the manuscript.

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
