# Peer review of "Clinical Application of Rapid Upper Limb Assessment and Nordic Musculoskeletal Questionnaire in Work-Related Musculoskeletal Disorders: A Bibliometric Study"

_ijerph, 2023, doi:10.3390/ijerph20031932_

Round 1
Reviewer 1 Report
Very interesting article and is valuable for general understanding of ergonomic risk factors in health care providers. the following are specific comments on the manuscript:
1. 1. Introductory paragraph is not focused. Needs work.
2. Line 81 – Responses are not validated, the questionnaire is.
3. Line 102 says that you searched for articles including BOTH RULA and NMQ but methods indicated that you selected 50 articles that contained RULA and 50 articles that contained NMQ. Need to make clear.
4. Flowchart is good.
5. First paragraph in Results is unclear.
6. What does length of article and number of citations have to do with value of article?
7. Length of time since publication may have more to do with number of citations than anything else.
8. Would be helpful if you included date of publication in table 1a.
9. Same for Table 1b.
10. Not entirely clear how the quartiles were decided for the journal rankings. Need to make the quartile designation clear before using in in the tables.
11. What is the significance of putting journals into quartiles?
12. Listing the primary author may be misleading when there are several. Might be more accurate to say first author.
13. Ranking institutions by countries is interesting but is it significant?
14. Identifying study type is of value but perhaps you need to briefly define what you mean by each study type. Some of those listed may overlap.
15. In paragraph ending at line 375, what subspeciality is it?
16. Good discussion of limitations.
Reviewer 2 Report
This study bibliometrically analyzed frequently cited papers on clinical applications of RULA and NMQ for assessing WMSDs risks in viewpoints of journal title, amount of citations, publication time, study type, institution of origin, level of evidence, quartile, contribution of primary authors, geographical distribution, and keywords. The number of paper citations of 50 or 106, which are the highest citation numbers for RULA and NMQ, respectively, cannot be said to be high enough to require some kind of analysis as in this manuscript. This may be because the survey area was limited to clinical applications of RULA and NMQ. So, the authors should clearly state why this study is needed and what meanings this study has. The authors have spent much time for preparing this manuscript, but the reviewer is wondering if this manuscript is worth publishing in this journal. It is strongly recommended to resubmit rewritten manuscript.
The followings need to be reconsidered or revised for making this paper better.
1. The reviewer is not native English-speaker, but it is recommended that the manuscript be reviewed by native English-speakers before submitting.
2. It should be stated why the study was limited to clinical applications of RULA and NMQ.
3. The numbers in the table 2, 3, 6 and 7 (< 7) are too small to make the analyses meaningful.
4. It should be explained that ‘level of evidence’ is.
Round 2
Reviewer 2 Report
There may be misunderstanding for the reviewer’s comments. The reviewer would like to ask the authors to carefully re-read first revieweing results and address them in the revised manuscript. It would be better to reflect the comments in the revised manuscript and resubmit it.
Reviewer’ comment: It should be stated why the study was limited to clinical applications of RULA and NMQ
Authors’ response: Thank you for your valuable comment.
The study was limited to clinical applications of RULA and NMQ only because these are the primary tools used to evaluate work-related musculoskeletal disorders among healthcare practitioners compared to other assessment tools.
Both these tools demonstrated greater reliability in evaluating the ergonomic hazards in various healthcare practitioners
Changes are incorporated in the Manuscript.
Page: 2, Line: 55-59
è The authors’ comments may be subjective and not based on objective research or data. Do the references of [4,13,14] show the reliability of RULA and NMQ as stated in the manuscript? Please check this. It is difficult for the reviewer to agree with the authors’ statement.
è The reviewer thinks that two things should be clearly stated for this manuscript to be published in this journal: First, why are the two methods selected among various assessing musculoskeletal risks? The two methods may have slightly different uses: RULA is used for assessing the size of musculoskeletal risks rather than the prevalence of WMSDs, while NMQ for investigating the prevalence of WMSDs. Second, why did the authors deal with the two methods’ applications to only clinical fields? Gómez-Galan et al. [15] presented that RULA have been applied to various knowledge categories.
Reviewer’s comment: The numbers in the table 2, 3, 6 and 7 (< 7) are too small to make the analyses meaningful.
Authors’ response: Thank you for your valuable comment.
Now the 10-point font size is kept for all the tables.
Changes are incorporated in the Manuscript as per the reviewer suggestion
è The reviewer does not mean that the number is font size, but the number of papers cited in the tables. Please re-read the reviewer’s first reviewing results.
Author Response
Thank you for your valuable comments.
Kindly find the attached file.
